# Can Increasing Scale Efficiency Curb Agricultural Nonpoint Source Pollution?

**DOI:** 10.3390/ijerph18168798

**Published:** 2021-08-20

**Authors:** Xiang Gao, Binglong Li, Song Jiang, Yunbin Nie

**Affiliations:** 1Department of Agricultural Economics, College of Economic and Management, China Agricultural University, 17 Qinghua East Road, Beijing 100083, China; gaoooxiang@cau.edu.cn (X.G.); nieyunbin@163.com (Y.N.); 2Department of Finance, School of Economics and Finance, Chongqing University of Technology, Chongqing 400054, China; jiangsong@cqut.edu.cn

**Keywords:** scale efficiency, agricultural NPS pollution, spatial effect, China

## Abstract

The Chinese government has made great efforts to improve the scale efficiency of land through various measures during recent years, hoping to realize the coordinated developing goal of promoting agricultural benefits and protecting the environment. Statistics show that China’s land scale efficiency has steadily increased, but agricultural nonpoint source (NPS) pollution has also increased, which seems contrary to the expected outcome. Can increasing scale efficiency really curb agricultural NPS pollution? This study uses provincial-level data from China, together with a panel model and spatial econometric model, to investigate the relationship between scale efficiency and agricultural NPS pollution. It is found that the increase of scale efficiency aggravates the agricultural NPS pollution, and the conclusion still holds after considering spatial effect. The results of spatial analysis shows that the agricultural NPS pollution is spatially dependent. Further decomposition of the spatial effect shows that the scale efficiency not only intensifies the local agricultural NPS pollution, but also has a spillover effect (though not statistically significant) on agricultural NPS pollution in the surrounding areas. It is worth noting that financial policy, raising wage income and upgrading industrial structure can effectively curb agricultural NPS pollution in this region and adjacent areas, which also deserves our attention in the control of agricultural NPS pollution. In addition, it is necessary to make financial and fiscal support policies specifically for the governance of agricultural NPS pollution, adjust the distorted prices of input factors such as chemicals and pesticide, and accelerate the transformation of small-sized farmers to family farms, in order to maximize the inhibitory effect of scale efficiency on relieving agricultural NPS pollution.

## 1. Introduction

Many developing countries are faced with low production efficiency and serious environmental pollution. In order to improve sustainable development in agriculture, many governments in developing countries have proposed to transform and upgrade their agricultural systems. For example, the Chinese government has proposed achieving high-quality development in agriculture many times, and the core content of high-quality development is to increase total factor productivity (TFP). TFP includes pure technical efficiency, scale efficiency and technological progress, among which scale efficiency is repeatedly mentioned in policy level and highly promoted in practice. The Chinese government has introduced a series of policies to encourage the farmers to expand the cultivated area, purchase trusteeship of agricultural production and participate in contract agriculture, increasing the scale efficiency steadily.

There are many ways to improve scale efficiency, and good results have been achieved in agricultural production practice. In China, the most common ways to increase scale efficiency are mainly divided into two categories. One type is driven by the transformation of land, such as encouraging farmers to transfer their land to large professional family farm and cooperatives. The other type is the agricultural production trusteeship, which provides farmers with the agricultural production service to realize the scale management at the level of service. In addition, there is also the rapid development of contract-driven agriculture. Relying on agricultural cooperatives, family farms and enterprises, adopting a pattern of integrating cooperative, production and farmers can expand contracted agricultural area. By the end of 2018, China’s land transfer rate reached 37%. In 2019, the country’s area of agricultural production trusteeships exceeded 100 million ha, serving an area of 573 million ha of food crops, and organizations of trusteeships reached 440,000. All these have promoted the improvement of agricultural scale efficiency, and played a positive role in expanding agricultural production efficiency, promoting farmers’ income increase and agricultural modernization.

At the same time, the improvement of scale efficiency brings some challenges and risks to the development of agriculture. Some agricultural operating entities blindly pursue expanding the planting area, ignoring scale efficiency, resulting in a waste of land resources and reduction of income. In addition, due to the expansion of farmers’ planting scale, the labor input under the traditional smallholder intensive farming production mode can no longer match the current land element level. What is more, urbanization has led to the outflow of rural labor. Farmers often adopt machinery and increase the input of chemicals to make up for the shortage of labor. The excessive use of agricultural machinery, especially chemical fertilizers and pesticides, has caused problems such as rising production costs and destruction of the ecological environment. Among these problems, agricultural nonpoint source (NPS) pollution is particularly prominent, which has serious consequences if not effectively controlled. Agricultural NPS pollution accounts for approximately 12 per cent of degraded land worldwide [1]. Therefore, it is of practical and theoretical significance to pay attention to the impact of scale efficiency on agricultural NPS pollution.

There are controversies about the relationship between scale efficiency and agricultural NPS pollution. On the one hand, increasing scale efficiency can curb agricultural NPS pollution [2,3,4]. The improvement of scale efficiency means that the allocation of production elements such as land, capital and labor inputs are more reasonable. The transformation of production pattern to green and modern will be promoted, which will reduce agricultural NPS pollution to a certain extent. For example, the trusteeship of agricultural production contributes to the reduced use of chemical fertilizers in China [5]. Through trusteeship of agricultural production, farmers can enjoy scientific and professional services provided by relevant agencies, including unified prevention and control, soil testing and formulated fertilization, and unified recycling of agricultural film, thereby reducing agricultural NPS pollution. On the other hand, scale efficiency may aggravate agricultural NPS pollution. The improvement of scale efficiency requires farmers to increase capital investment, such as renting land and purchasing trusteeship of agricultural production. In order to recover costs and obtain profits as soon as possible, farmers are very easy to adopt short-sighted production behaviors to raise yield, such as increasing the input of fertilizers and pesticides. Studies in Ethiopia and Ghana have also shown that trusteeship of agricultural production increases the intensity of fertilizer use [6,7,8]. What is more, excessive use of fertilizers and pesticides is the most important driver of agricultural NPS pollution in China [9,10]. Can the improvement of scale efficiency inhibit agricultural NPS pollution in China? Does agricultural NPS pollution show a spatial effect? This study attempts to answer the above questions.

The main contributions of this study are as follows: (1) This study analyzes the relationship between scale and agricultural NPS pollution from the efficiency level. In previous studies, when discussing the relationship between the above, planting land scale or service scale was often used. However, expanding scale does not mean improving the efficiency of resource allocation. Scale efficiency can be closer to the actual and core connotation of agricultural production, and it is more accurate and representative. (2) Existing studies mostly regard different research areas as independent individuals, ignoring the spatial spillover possibility of agricultural NPS pollution. This study considers the spatial correlation and interaction characteristics of agricultural NPS pollution among provinces, and uses spatial econometric models to test and estimate its spatial effects.

The other parts of this study are as follows: The second part reviews the relevant literature; the third part is the research method; the fourth part includes variables and data; the fifth part is the results and discussion; the sixth part is a summary of the full text.

## 2. Literature Review

The concept of nonpoint source pollution originated from the Clean Water Act (CWA) amendment in the United States. Although the concept of NPS pollution was put forward earlier in the world, it was not until the 1960s that comprehensive understanding and research were carried out successively [11]. With the continuous advancement of research, the definition of agricultural NPS pollution is gradually improved and clear. From the perspective of agricultural production, agricultural NPS pollution refers to the water environment pollution formed by the surface runoff from farmland and the leakage of nutrients such as nitrogen and phosphorus, pesticides, and other pollutants in agricultural production activities [12]. Based on the analysis of the formation process of agricultural NPS pollution, it means that dissolved pollutants and solid pollutants from non-specific locations are absorbed into the receiving water body through runoff process under the scouring action of precipitation or melting snow [13].

Agricultural NPS pollution is the negative environmental output of the agricultural production system, which has the characteristics of large randomness in the formation process, many influencing factors, wide distribution, long latent period and great harm. If left untreated, it will have many undesirable consequences for the natural environment and economic development, such as polluting the water environment and soil, endangering the quality and safety of food, and declining economic returns. By the end of the twentieth century, about 30% to 50% of surface water or water bodies worldwide had been affected by NPS pollution [14]. The survey results of the Ministry of Health of China showed that in the first quarter of 2004, the rate of high-toxicity and high-residue banned pesticides in leafy vegetables nationwide was still 10.5%, and more than 90% of the food on the table of urban residents had been directly or indirectly contaminated. Therefore, it is particularly urgent and important to control agricultural NPS pollution.

To combat agricultural NPS pollution, governments around the world have adopted a combination of legal, economic, technological and voluntary public participation measures. Regulatory means are one of the most effective measures against agricultural NPS pollution in various countries. The United States first stipulated the treatment of NPS pollution in the Clean Water Act of 1972. The Federal Water Pollution Control Act (FWCA) proposed sustainable agricultural practices (SAPs). SAPs are environmental management strategies that take scientific measures to reduce pollution when maximized crop output is obtained. At the economic level, “Pigou means” and “Coase means” are mainly used to control it. Taxes, subsidies, administrative intervention and so on constitute the main content of “Pigou means.” Previous studies have proposed improving management of agricultural NPS pollution by means of tax policy, subsidy policy and administrative intervention [15,16,17]. Tools such as sewage charges and carbon emission rights trading constitute the main content of the “Coase means.” Horan and Shortle conducted a quantitative study on emissions trading between NPS pollution and point source pollution [18]. In terms of technical control measures, the adoption of alternative technologies with low cost and strong operability plays a positive role in the treatment of agricultural NPS pollution. The United States encourages farmers to adopt environmentally friendly techniques in their agricultural practices, such as soil testing, crop rotation, riparian buffers and vegetation filters. The EU promotes ecological agricultural technology and bans the use of chemical fertilizers and pesticides in agricultural production. At the level of public participation, the public is encouraged to take the initiative to participate in various environmental actions. Practice has proved that public participation in environmental management is a flexible and effective control measure, which can play an auxiliary role in the implementation of other control measures [13].

In order to deal with agricultural NPS pollution more effectively, it is necessary to identify the sources of NPS pollution and the factors controlling NPS pollution. There are factors controlling agricultural NPS pollution, at both macro and micro level scales. At the macro level, factors such as market, FinTech, economic development level, agricultural land endowment and subsidy policies cannot be ignored. Labor market and fertilizer price policies affect the level of fertilizer application, which has an effect on agricultural NPS pollution emissions [19,20]. Jiang et al. conducted research from a technical perspective, and believed that FinTech development can curb agricultural NPS pollution [21]. Schreinemachers and Tipraqsa studied the influence mechanism of agricultural land endowment on agricultural NPS pollution, and found that abundant agricultural land endowment helps to reduce the intensity of chemical fertilizer input, thus reducing NPS pollution [22]. Kurkalova et al. found that financial incentives, such as paying extra insurance fees for farmers, can promote farmers to take protective farmland protection measures, thus helping to reduce agricultural NPS pollution [23]. At the micro level, agricultural NPS pollution is affected by the education level of farmers, family income, land scale, difficulty in obtaining environmental protection technology, agricultural production experience, awareness of pollution, agricultural technology training and agricultural cooperation pattern [24,25,26].

There are three viewpoints on the relationship between scale efficiency and agricultural NPS pollution. The first is that scale can effectively curb the level of agricultural NPS pollution. A large number of studies based on China have shown that there is a negative correlation between scale and agricultural NPS pollution [2,27,28]. The second is that the improvement of scale efficiency will aggravate the degree of agricultural NPS pollution [3]. Similar findings have been found in studies of other developing countries [29,30,31]. What is more, other studies have found that there is no significant promoting or inhibiting relationship between the two [32].

However, studies to date have not reached a consensus on the relationship between scale efficiency and agricultural NPS pollution. Second, researches on scale and agricultural NPS pollution are relatively lacking, especially those from the perspective of efficiency. Third, most previous studies have ignored the spatial correlation.

## 3. Research Methods

### 3.1. Basic Model Setting

Previous studies on environmental impact have revealed its main influencing factors, which can be traced back to the IPAT model proposed by Ehrlich and Holdren [33]. This model is mainly used to measure the relationship between environmental Impact(I) and Population(P), Affluence(A) and Technology(T) with the following expression:(1)I=PAT

In Equation (1), environmental changes are mainly driven by three factors: population, economy development and technological level. It is impossible for a single factor to act on the environment independently of other factors. The theory also points out that population growth and human activities have a negative impact on the environment, and they vary with the level of economic and technological development. Environmental Kuznets curve (EKC) reveals a “U” shape between environmental pollution and economic development. Similarly, in agricultural production, the activities of each person will have an impact on the environment. At the same time, environmental impact will change with different levels of income and agricultural management techniques. The improvement of scale efficiency helps farmers to obtain benefits of scale economy, which can increase their income and reduce its negative impact on the environment. What is more, scale efficiency is actually determined by the production and management activities of farmers, and it represents the level of agricultural production and management technology, which in turn has an impact on environmental pollution. Therefore, the theoretical connection between scale efficiency and agricultural NPS pollution is established. Accordingly, we set the basic model as follows:(2)$$NPS_{i,t}=a_{0}+a_{1}Se_{i,t}+\lambda Control_{i,t}+\mu_{i,t}$$

In Equation (2), NPS is the level of agricultural NPS pollution, Se is scale efficiency and Control represents other variables which can affect the agricultural NPS pollution. $a_{0}$ is constant term. $a_{1}$ and λ are parameter terms to be estimated. μ is the random error term.

Studies reveal that agricultural NPS pollution can be also affected by factors such as fiscal policies, financial policies, wage, urbanization and industrial structure. The “Pigou means” is a common tool to control environmental pollution, and the fiscal policy is the main component of it. Ribaudo and Griesinger et al. both advocated the implementation of fiscal policies to guide farmers to adopt green production technologies, so as to achieve the goal of reducing NPS pollution at source [16,17]. Financial policies can reduce the financing costs of production entities [34], and provide financial support for their adoption of green production techniques and other environmentally friendly behaviors [35]. Wage also affects agricultural NPS pollution. In the IPAT model, affluence is closely related to environmental impact [33], and the EKC also reveals the inverted “U-shaped” relationship between environmental pollution and the economic development. Therefore, agricultural NPS pollution will change with the increase of wage. Urbanization is one of the most important changes in Chinese society over the past 40 years, which has a far-reaching impact on agricultural development and the environment. The shortage of labor factors caused by urbanization has increased the amount and extent of fertilizer application [36], aggravating agricultural NPS pollution. The relationship between industrial structure and environment is inseparable [37,38]. The influence of different industrial structure on environment is different, and a reasonable adjustment of the industrial structure is crucial to environmental protection. In order to make the model more reasonable, we introduced the above 5 variables into model (2) and expand it into model (3):(3)$$NPS_{i,t}=\beta_{0}+\beta_{1}Se_{i,t}+\beta_{2}Fin_{i,t}+\beta_{3}Pr_{i,t}+\beta_{4}W_{i,t}+\beta_{5}Urb_{i,t}+\beta_{6}Str_{i,t}+\mu_{i,t}$$

In Equation (3), $NPS_{i,t}$ is the level of agricultural NPS pollution, i represents the province and t represents the year. $Se_{i,t}$, $Fin_{i,t}$, $Pr_{i,t}$, $W_{i,t}$, $Urb_{i,t}\;$and $Str_{i,t}$ represent the scale efficiency, financial policy, fiscal policy, wage, urbanization and industrial structure, respectively. $\beta_{0}$ to $\beta_{6}$ are constant term and parameter terms to be estimated. $\mu_{i,t}\;$is the random error term.

### 3.2. Setting of Spatial Panel Model

There may be differences in agricultural NPS pollution levels among different regions, and agricultural NPS pollution is mostly carried by water. In addition, the similar geomorphology, hydrological characteristics and climatic conditions between neighboring provinces make the spatial transfer of agricultural NPS pollution possible. Therefore, we need to consider the possible spatial spillover effects of agricultural NPS pollution. In order to confirm whether agricultural NPS pollution has spatial correlation, this study uses the Global Moran’s I index to conduct spatial autocorrelation test, and the expression is as follows:(4)$$Moran^{\prime }s\;I=\frac{\sum_{i=1}^{n}\sum_{i=1}^{n}w_{ij}\left(x_{i}-\bar{x}\right)\left(x_{j}-\bar{x}\right)}{S^{2}\sum_{i=1}^{n}\sum_{j=1}^{n}w_{ij}}$$
(5)$$S^{2}=\frac{\sum_{i=1}^{n}{\left(x_{i}-\bar{x}\right)}^{2}}{n}$$

In Equations (4) and (5), $x_{i}$ and$\;x_{j}$ represent the agricultural NPS pollution, i and j represent the region. $w_{ij}$ represents the adjacent relationship between the two regions. n denotes the quantity of different areas and $S^{2}\;$represents the variance of regional agricultural NPS pollution.

Agricultural NPS pollution in adjacent areas may influence each other and eventually form a balanced result. Accordingly, we improved Equation (3) to establish a spatial autoregressive model (SAR). The expression is shown in Equation (6):(6)$$NPS_{i,t}=\beta_{0}+\rho WNPS_{i,t}+\beta_{1}Se_{i,t}+\beta_{2}Fin_{i,t}+\beta_{3}Pr_{i,t}+\beta_{4}W_{i,t}+\beta_{5}Urb_{i,t}+\beta_{6}Str_{i,t}+\varepsilon_{i,t}$$

Among them, W is the spatial weight matrix, $WNPS_{i,t}$ represents the degree of spatial dependence of agricultural NPS pollution. *ρ*, $\beta_{0}$*~*$\beta_{6}\;$respectively denote the parameters to be estimated, and ε is a random interference term. The other variables have the same meanings as Formula (3).

## 4. Variables and Data

### 4.1. Variable Description

#### 4.1.1. Dependent Variables

The dependent variable in this study is agricultural NPS pollution. The common agricultural NPS pollution refers to the rural environmental pollution caused by the unreasonable use of chemical fertilizers, pesticides, agricultural film, feed and veterinary drugs in agricultural production, as well as the untimely or improper treatment of livestock and poultry manure, crop straw, agricultural film domestic garbage and domestic sewage. What needs to be explained here is that the pollution caused by livestock and poultry farming waste, rural domestic sewage and domestic solid waste is quite different from the planting industry. In order to make the research more pertinent, this study focuses on the agricultural NPS pollution in the production of planting industry.

Agricultural NPS pollution mainly refers to the pollution of water, soil, air and agricultural products in the rural distracts caused by the unreasonable treatment of chemical substances such as fertilizers, pesticides, and mulching films used by farmers in agricultural production [1]. In this study, the main sources of agricultural NPS pollution are chemical fertilizer, pesticide, agricultural plastic film and diesel oil used in agricultural machinery. No matter whether these inputs are absorbed by crops or are overused, they often pollute water and soil under the multiple effects of rainfall, sediment and irrigation, and become the source of agricultural NPS pollution. In addition, because agricultural NPS pollution is scattered, concealed, random and difficult to monitor, the measurement of agricultural NPS pollution in the past was more limited to the single source of pollution. This can only reflect one aspect of agricultural NPS pollution, which is one-sided. In order to carry out a more comprehensive measurement of agricultural NPS pollution, this study adopted the unit survey method, and divided it into four units: fertilizer, pesticide, agricultural films and diesel according to the main pollution sources of agricultural NPS pollution. At the same time, drawing on the weight processing method of the United Nations human development index (HDI) and economic vulnerability index (EVI), a simple and transparent weight setting method was adopted. That is, the weight of fertilizer (CF), pesticide (PE), agricultural mulch (MF) and diesel usage amount (DF) is set to 0.25. And the level of agricultural NPS pollution in China can be expressed as Equation (7):(7)$$NPS_{i,t}=0.25\times CF_{i,t}+0.25\times PE_{i,t}+0.25\times MF_{i,t}+0.25\times DF_{i,t}$$

#### 4.1.2. Independent Variables

Scale efficiency (Se): The scale efficiency in this study is mainly measured by scale efficiency change index (SECH). The Chinese government has continuously encouraged farmers to transfer land, purchase trusteeship of agricultural production and participate contract agriculture. However, whether it is expanding the area of arable land, purchasing trusteeship of agricultural production or participating in contract farming, the logic behind it is to improve the scale efficiency of agriculture. Therefore, the scale efficiency is closer to the reality of agricultural production. In addition, scale efficiency can accurately measure the implementation effect of various ways, thus making the measurement indicators more accurate, scientific and representative. In order to measure the SECH, this study used two steps. One is to set input and output variables. Agricultural added value is set as the output variable, and the sown area of crops and rural population are set as the input variable. Second, DEA-Malmquist method is used to measure the total factor productivity change index (TFPCH) and is decomposed, which is shown in Equation (8).
(8)$$\begin{array}{c} \underset{TFPCH}{\underbrace{M\left(x^{t+1},y^{t+1},x^{t},y^{t}\right)}}={\left[\frac{D_{0}^{t}\left(x^{t+1},y^{t+1}\right)}{D_{0}^{t}\left(x^{t},y^{t}\right)}\times \frac{D_{0}^{t+1}\left(x^{t+1},y^{t+1}\right)}{D_{0}^{t+1}\left(x^{t},y^{t}\right)}\right]}^{\frac{1}{2}}\\ =\underset{\underset{=SECH\times PECH}{\underbrace{EFFCH}}}{\underbrace{\frac{D_{0}^{t+1}\left(x^{t+1},y^{t+1}\right)}{D_{0}^{t}\left(x^{t},y^{t}\right)}}}\times \underset{TECHCH}{\underbrace{{\left[\frac{D_{0}^{t}\left(x^{t+1},y^{t+1}\right)}{D_{0}^{t+1}\left(x^{t+1},y^{t+1}\right)}\times \frac{D_{0}^{t}\left(x^{t},y^{t}\right)}{D_{0}^{t+1}\left(x^{t},y^{t}\right)}\right]}^{\frac{1}{2}}}} \end{array}$$

Financial policy (Fin): Financial policy is an important means to solve agricultural NPS pollution, and funds to support agriculture can promote the green development of agriculture. Financial support for agriculture can reduce the financing costs of operating entities and thus promote green investment in agriculture, especially green credit, which has a moderating effect on market failure [34]. However, China has not separately counted and listed the scale of green credit and green bonds presently. Therefore, this study uses the total agricultural credit scale to measure the strength of financial policy.

Fiscal policy (Fis): Policies and financial investment related to environmental governance can reduce pollutant emissions [39]. Agricultural NPS pollution has increasingly become focus of many governments. In order to alleviate agricultural NPS pollution, the state has issued a series of fiscal policies. For example, the Chinese government has implemented subsidies and tax exemptions for selling organic fertilizer products and using fewer chemical fertilizers and pesticides these years. These fiscal policies mainly include price subsidies and tax exemptions, and expenditures on agriculture, forestry and water supplies are an important means for the Chinese government to support the green development of agriculture. Therefore, this study uses local agriculture, forestry and water affairs expenditures to measure the intensity of fiscal policy to support for agriculture.

Wage (W): Wage affects the level of agricultural NPS pollution. The level of economic development directly determines the farmers’ pattern of production, management capabilities and environmental awareness, thus affecting agricultural NPS pollution. From the EKC, it can be seen that there is an inverted “U-shaped” relationship between country’s economic development level and environmental pollution. When the level of economic development is low in the early stage, agricultural NPS pollution increases with the promotion of residents’ wage, and eases with the increase of wage after reaching the critical point. The adoption cost of environmentally friendly technologies is relatively high, and the level of wage determines whether the farmer is able to bear the cost to some extent. This study uses the disposable income of rural residents to measure the farmers’ wage.

Urbanization (Urb): There is a link between urbanization and agricultural NPS pollution that cannot be ignored. The off-farm shift of labor force brought about by urbanization has increased the opportunity cost of labor. In addition, the relative changes in the prices of agricultural factors have driven farmers to choose more labor-saving and farmland-saving technologies such as chemical fertilizers and pesticides. Additionally, due to farmers’ awareness of production risk avoidance [40] and lack of scientific fertilization knowledge and technology, it is also easy to cause excessive use of chemical fertilizers and pesticides, which intensifies agricultural NPS pollution. At the same time, the development of urbanization will increase the income and education level of migrant rural workers. On the one hand, it is helpful to improve the awareness of green production of farmers. On the other hand, it can provide financial support for the adoption of environmentally friendly technologies such as soil testing and formula fertilization technology, and alleviate agricultural NPS pollution. However, whether urbanization will curb or aggravate agricultural NPS pollution remains to be tested. This study uses the ratio of urban population to total population to measure urbanization.

Industrial structure (Str): The industrial structure affects the level of agricultural NPS pollution. Generally speaking, extensive industrial structure tends to gain economic benefits at the expense of the environment, and moving towards resource-saving and environment-friendly industrial structure will have a positive impact on environment. Therefore, the transformation of industrial structure can improve environmental pollution to a certain extent. This study uses the ratio of added value of agriculture to sum of the added value of the secondary and tertiary industries to measure changes in the industrial structure.

### 4.2. Data Sources and Description

The data used in this study are panel data of 31 provinces in the mainland of China from 1999 to 2018. These data are all collected from official databases in China such as the China Statistical Yearbook, China Rural Statistical Yearbook, China Financial Yearbook, and Compilation of Statistical Data for the Sixty Years of New China: 1949–2008. In addition, this study replaced and supplemented missing data with the average value of the two years before and after.

As shown in Table 1, the standard deviation of agricultural NPS pollution in all provinces is large, which is 188.928. The maximum value is 859.86, the minimum value is 3.221, and the average value is 240.134, indicating that China’s agricultural NPS pollution levels are relatively high. In addition, there are big differences between different provinces. Considering that there are obvious differences in agricultural NPS pollution among different provinces, and due to natural, economic and social conditions, they may influence each other to reach the final equilibrium state. Therefore, this study will also choose an appropriate spatial model for further discussion.

The standard deviation of scale efficiency is 0.814, and the difference between the maximum value of 7.777 and the minimum value of 0.107 is more than 70 times. This indicates that there is a gap in the scale efficiency of various regions in China, which may be an important reason of agricultural NPS pollution. In terms of financial policies and fiscal policies, their standard deviations are 4.90 × 10^7^ and 259.742, respectively. This means that different provinces and regions have significant differences in financial policies and fiscal policies for supporting agricultural development. The standard deviation of the wage is 5048.142 yuan. Judging from the difference between the maximum value of 30,374.73 yuan and the minimum value of 1258 yuan, there is a large imbalance in average wage of farmers in various provinces. The province with the highest urbanization rate has reached 89.6%, while the lowest province is only 22%. The national average is 49%, and its standard deviation is 0.152. The gap of urbanization among provinces still exists. The average value of the industrial structure is 0.087, which indicates that there is huge space for upgrading agricultural industrial structure from the national average level.

## 5. Results and Discussion

### 5.1. Results of Panel Model

In order to explore the relationship between scale efficiency and agricultural NPS pollution level, this study first applied the panel model to estimate it based on the collected panel data, and reported the results of mixed effect model, fixed effect model and random effect model in Table 2, respectively. In addition, the F test (Prob > F = 0.0000,) showed that the null hypothesis was rejected. In other words, the fixed effects regression model was significantly better than the mixed regression model. The results of Hausman test (Prob > chi2 = 0.0005) indicated that the fixed effects regression model was significantly better than the random effects regression model. Therefore, this section mainly explains the regression results of the fixed effects model.

The regression results of fixed effect model show that the effect of scale efficiency on agricultural NPS pollution is positive, but the significance test is not passed. This means the improvement of scale efficiency aggravates the degree of agricultural NPS pollution. The possible reasons are the following three aspects. First, the Chinese government still emphasizes on “food security” at the strategic level and requires storing grain in the ground, which means giving priority to output. In addition, the rapid development of industry in recent years has greatly reduced the prices of chemical fertilizers, pesticides and other production factors, gradually becoming the best choice for agricultural producers to increase output. However, excessive use of input elements such as fertilizers and pesticides will lead to an increase in undesired output and limit the room for efficiency improvement. Second, farmers at the micro level lack the enthusiasm for adopting green production technologies. It is well known that although the adoption of green production technology has positive externalities for the environment, its short-term return on investment is low, and most farmers pursue short-term profit maximization. In order to obtain more profits and avoid yield risks, farmers will choose to use excessive chemicals, thereby exacerbating agricultural NPS pollution. Third, although the government has encouraged land transfer and trusteeship of agriculture production to improve scale efficiency, the number of small farmers in China accounted for more than 98% of agricultural business entities according to the data of the third national agricultural census. Among them, small farmers account for 90% of the agricultural employees, and the cultivated land operated by small farmers accounts for 70% of the total cultivated land area. Due to China’s national conditions, small and medium-sized farmers will still be the main body in the present and future for a long period of time. These farmers have little room for improvement in scale efficiency, and may not be able to leverage the levers of their agricultural green leading functions, which will also limit scale efficiency to improve the mitigation effect on agricultural NPS pollution. In addition, the reason why scale efficiency did not pass the significance test may be that the spatial effect of agricultural NPS pollution was not taken into account, leading to biased regression results. In the next part, we will explore this.

According to the regression results of control variables, financial policy can significantly inhibit agricultural NPS pollution, which is mainly related to the rapid development of green finance in China. Green finance aims to regulate the capital supply structure, improve the green development space of agriculture and other industries, and guide the environment-friendly development pattern [41], which is conducive to the realization of green development of agriculture. The effect of fiscal policy on agricultural NPS pollution is significantly positive. Although financial support for agriculture has increased year by year in recent years, the lack of funds directly targeting agricultural NPS pollution, especially inadequate investment in agricultural ecological infrastructure, is not conducive to the effective control of agricultural NPS pollution. Wage can effectively reduce agricultural NPS pollution level. With the increase of wage, it can effectively stimulate farmers’ awareness of environmental protection and promote their adoption of green production behavior, thus reducing agricultural NPS pollution. Urbanization has exacerbated the level of agricultural NPS pollution. On the one hand, urbanization has aggravated agricultural NPS pollution by changing land use patterns. The demand for urban construction land brought by urbanization has occupied a large amount of arable land. In order to ensure agricultural output and profit, it is necessary to increase the input of chemical fertilizers and pesticides and other factors, which intensifies agricultural NPS pollution. On the other hand, the transfer of off-farm labor brought about by urbanization has increased the degree of agricultural NPS pollution. Urbanization leads to the increase of opportunity cost of agricultural labor, which induces farmers to adopt labor-saving production technology to make up for the shortage of labor. Chemical fertilizer and pesticide become the best choice because of their low cost. The effect of industrial structure on agricultural NPS pollution is negative. The gradual upgrading from extensive industrial structure is beneficial to the reduction of agricultural NPS pollution.

### 5.2. Estimation Results of the Spatial Effect

The premise of using spatial model is the existence of spatial dependence. Therefore, this study uses Moran’s I index mentioned above to test the spatial correlation. The results show that Moran’s I index is all positive and passed the significance test at the 1% level except for a few years. The results indicate that agricultural NPS pollution is spatially dependent, and it is necessary to use spatial models to estimate spatial effects. In this study, we used spatial autoregression (SAR) to analyze the spatial effects.

Table 3 is the regression results of considering the inter-provincial spatial interaction effects of agricultural NPS pollution. At the same time, in order to eliminate the problem of multicollinearity that may exist in the model, this study adopted the method of gradually introducing explanatory variables into the model during regression. As if shown in regression results, it can be found that the coefficients ρ of model (4) to model (9) are all significantly positive, indicating that agricultural NPS pollution has an inter-provincial spatial interaction effect. The coefficient signs and significance levels of explanatory variables in the six models are basically the same, meaning that the empirical results are robust to a certain extent. What is more, considering the R^2^ of Model (9) is the largest and the explanatory power is also the strongest after the explanatory variables are gradually added, so this study mainly focused on regression results based on model (9).

It can be seen from model (9) that the coefficient of ρ is 0.337, and it has passed the significance test at 1%. The results show that agricultural NPS pollution in one province is affected by the level of agricultural NPS pollution in neighboring provinces. The possible reasons are as follows: First, from the perspective of natural endowments, similar climatic, geomorphological and hydrological characteristics between neighboring provinces provide the possibility for spatial diffusion of agricultural NPS pollution. Agricultural NPS pollution is mostly carried by water, and it can enter the water body through diffusion, confluence and diversion. For example, rainfall can bring agricultural NPS pollution sources to rivers and lakes in neighboring provinces, so that agricultural NPS pollution can realize spatial transfer. Secondly, from the perspective of environmental regulation conditions, different provinces have differences in the ability and intensity of environmental supervision, which provides institutional conditions for the spatial spillover of agricultural NPS pollution. Neighboring provinces have convenient transportation and similar agricultural infrastructure, and agricultural production entities tend to seek advantages and avoid disadvantages. When facing relatively strict environmental regulations, it is easy for them to choose to transfer agricultural industries to nearby areas with smaller environmental regulations. Finally, the demonstration of agricultural production activities in adjacent provinces strengthens the spatial interaction of agricultural NPS pollution. Agricultural production activities in one area have a strong demonstration effect on neighboring areas. For example, the spillover of agricultural land use mode, intensity and technology helps to form the spatial agglomeration of factors, thus affecting the agricultural production activities and NPS pollution emission intensity in the neighborhood.

From the regression results of core explanatory variables, the effect of scale efficiency on agricultural NPS pollution is positive, and the estimated coefficient also becomes significant. This shows that, when discussing the relationship between scale efficiency and agricultural NPS pollution, it is necessary to consider its spatial effect. In terms of control variables, after considering the spatial interaction effect, the estimated results of financial policy, fiscal policy, wage level, urbanization and industrial structure are consistent with the above part. Therefore, the reasons are not discussed in this section.

### 5.3. Estimated Results of Decomposed Spatial Effect

In order to further measure the complex spatial correlation between adjacent areas, it is often necessary to split the spatial effects when using spatial models. Lesage et al. [42] proposed that direct effect and indirect effect can be used to describe the spatial marginal effect of adjacent areas, and the coefficient can be obtained by calculating partial derivatives. In this study, the direct effect refers to the influence of explanatory variables in local province on agricultural NPS pollution, and the indirect effect refers to the influence of explanatory variables in the neighboring province on agricultural NPS pollution. The total effect obtained is equal to the sum of direct effect and indirect effect. Among them, the direct effect includes two influence paths. The first path refers to the direct impact of the explanatory variables on local area’s agricultural NPS pollution. The second is a feedback path. When explanatory variables in local province change the level of agricultural NPS pollution in the neighboring regions by affecting the explained variables in the neighboring regions, the change of agricultural NPS pollution will eventually feedback back to the local region. Indirect effects also include two paths. One refers to the influence of explanatory variables in adjacent provinces on agricultural NPS pollution in local province. The other refers to the influence of explanatory variables in neighboring provinces on agricultural NPS pollution in the neighboring provinces, which in turn affects the agricultural NPS pollution in local province.

As shown in Table 4, the spatial decomposition effect results of driving factors on agricultural NPS pollution are presented. The direct and indirect effects of scale efficiency on agricultural NPS pollution are both positive, indicating that the improvement of scale efficiency will aggravate the degree of agricultural NPS pollution. At the initial stage, the improvement of scale efficiency is helpful to the reasonable allocation of various input elements, especially to make the application of chemicals more reasonable, thus reducing agricultural NPS pollution. However, with the improvement of scale efficiency, it is easy to face the problems of labor shortage and high cost of local labor. In this situation, it is natural to produce the “substitution effect” of agricultural chemicals to make up for the labor shortage. In addition, profit maximization is still the primary goal of most agricultural business entities. Improving scale efficiency requires capital investment. In order to recover costs as soon as possible and avoid production risks, intensive application of chemicals may be used by farmers to ensure the increase of production. This will cause the chemical “reduction effect” at the initial stage to be gradually offset, and further aggravate the local agricultural NPS pollution. What is more, the indirect spillover effect of scale efficiency is not significant. This is mainly due to the differences in the ways and policy contents of improving scale efficiency among neighboring provinces, and the transformation cannot be achieved overnight.

The direct and indirect effects of financial policies on agricultural NPS pollution are negative, and both of them pass the significance test of 1%. Financial policy can reduce agricultural NPS pollution mainly because of the rapid development of green finance in China in recent years. Green finance can play a synergistic effect with existing environmental regulations, and guide the upgrading of agricultural green production technology to reduce the agricultural NPS pollution in local region. The indirect effect is negative, which indicates that financial policy has spatial spillover effect. Financial policies guide the upgrading of local agricultural green production technology, which can be spread to adjacent areas, thereby helping to alleviate the agricultural NPS pollution in surrounding areas.

The direct and indirect effects of fiscal policy on agricultural NPS pollution are significantly positive. First of all, fiscal policies have increased the intensity of the use of agricultural machinery. For example, farm machinery purchase subsidy encourages farmers to use large agricultural machinery, which promotes the use of fossil fuels and aggravates agricultural NPS pollution in local region. Second, although there are special subsidies linked to grain planting areas, such as direct grain subsidies and producer subsidies, the income of cash crops is significantly higher than that of grain crops. Therefore, the trend of non-grain-oriented planting structure has become gradually prominent. Although cash crops have more benefits, the pollution generated during the production process is generally higher than grain crops, especially in the use of chemical fertilizers [43]. The transformation of planting structure will aggravate the agricultural NPS pollution in local region. Finally, the existing fiscal policies for direct treatment of agricultural NPS pollution are insufficient, especially the lack of public investment, which also leads to the failure of fiscal policies to achieve the established goals. The indirect effect is positive, indicating that local fiscal policy has aggravated the level of agricultural NPS pollution in neighboring provinces. The agricultural industry in the adjacent area is demonstrative, and the transformation of local agricultural production activities caused by fiscal policies has a demonstrative and leading role in surrounding areas. For example, the surrounding area and the local area are consistent in the non-grain-oriented planting structure, which will increase the degree of agricultural NPS pollution in the neighborhood.

The direct and indirect effects of wage on agricultural NPS pollution are both significantly negative. Increasing wage is helpful to stimulate the environmental awareness of agricultural production entities, and realize the transformation from traditional production mode to green production mode, thereby reducing the intensity of agricultural NPS pollution. In addition, small farmers are still the main operating entities of agricultural production in China. Compared with new operating entities, small farmers’ cost of adopting environmentally friendly technologies is higher, and the increase in wage can provide financial support for the adoption of such technologies. The indirect effect is negative, meaning that an increase in wage of farmers in local province can reduce agricultural NPS pollution in neighboring provinces. If there is a wage gap between neighboring provinces, the labor force will flow from the “income depression” to the “income plateau” and eventually reach an equilibrium state. As long as the wage gap exists, this phenomenon of labor immigration will occur. The inter-provincial immigration of labor has two benefits. On the one hand, it can increase the wages and make farmers in surrounding provinces less sensitive to the cost of adopting environmentally friendly technologies. On the other hand, farmers coming from adjacent backward provinces working in developed provinces can broaden their horizons, contact the new green development concept and practice it in actual agricultural production, thereby reducing the agricultural NPS pollution in neighboring provinces.

The direct and indirect effects of urbanization on agricultural NPS pollution are both significantly positive. Urbanization has changed the way of land use and affected agricultural production activities. Urbanization has provided a large number of local farmers with off-farm employment opportunities, resulting in the reduction of time spent in local agricultural labor. Therefore, it is easy for farmers to choose the former between chemical fertilizer which is time-saving and labor-saving, and organic fertilizer, which is time-consuming and labor-consuming. Chemical fertilizer can increase agricultural NPS pollution in local province. The indirect effect is positive, indicating that the advancement of urbanization is conducive to the inter-provincial transfer of agricultural NPS pollution. For the developed provinces, urbanization largely benefits from the cross-regional optimal allocation of labor from the backward provinces. As a result, the labor cost of agricultural production in backward areas continues to rise. Driven by profit maximization, farmers in neighboring provinces are more inclined to choose low-cost fertilizers to replace labor with high costs, which intensifies the degree of agricultural NPS pollution in neighboring provinces.

The direct and indirect effects of industrial structure on agricultural NPS pollution level are both negative and pass the significance test at 5%. It means that industrial structure is an important factor affecting agricultural NPS pollution. The adjustment of industrial structure means mutual integration and optimal allocation of resources between industries, which promotes the transformation of agriculture from a resource-consuming pattern with high energy consumption and low output to a resource-saving pattern. Such transformation can reduce the use of chemicals such as pesticides and fertilizers at the source, and thus alleviate agricultural NPS pollution in local provinces. The indirect effect is negative, indicating that industrial structure has a demonstration effect among neighboring provinces. In addition, the spillover effect of the demonstration effect is obvious, which can effectively reduce the agricultural NPS pollution in neighboring provinces.

## 6. Conclusions

From the perspective of efficiency, this study constructed a theoretical model of the impact of scale efficiency on agricultural NPS pollution. Based on the data collected of different provinces in China, the panel fixed effects regression model was firstly used to estimate the impact. Then, considering the spatial correlation of agricultural NPS pollution, the spatial econometric model was used. Finally, the spatial effect is further decomposed into direct effect and indirect effect for specific analysis. The research results of this study mainly include the following aspects:

(1). The improvement of scale efficiency intensifies the degree of agricultural NPS pollution. As the Chinese government currently emphasizes “food security” at the strategic level, it is required to ensure the necessary grain planting area and stable grain production. In addition, the farmers pursue maximizing profit and avoid production risk, resulting in the “substitution effect” of chemicals to offset the “rational allocation effect” of input elements brought about by improving scale efficiency. Gathering these reasons, the improvement of scale efficiency has not been able to effectively suppress agricultural NPS pollution.

(2). Agricultural NPS pollution has spatial correlation. Agricultural NPS pollution can be transferred across provinces due to their similar natural conditions such as climate and terrain, differences in environmental regulations and demonstration of agricultural production. In other words, the increase of agricultural NPS pollution level in one province will lead to the increase of agricultural NPS pollution intensity in the surrounding areas.

(3). The scale efficiency of this region not only aggravates the degree of local agricultural NPS pollution, but also has a spillover effect on agricultural NPS pollution in neighboring area, but this spillover effect is not significant.

(4). Financial policies, raising wage and upgrading of industrial structure can effectively restrain agricultural NPS pollution in this area and adjacent areas, which also deserves our attention in reducing agricultural NPS pollution.

In summary, improving scale efficiency is an inevitable requirement for the realization of agricultural transformation and green development. It can not only improve the allocation efficiency of agricultural production factors, but also help to increase agricultural income. At the same time, agricultural NPS pollution is one of the important problems faced by many developing countries. Fortunately, some studies have begun to focus on the promotion effect of scale efficiency on green agricultural development, but whether the goal of curbing agricultural NPS pollution can be achieved by improving scale efficiency is still controversial. Therefore, this study constructs a theoretical framework for the impact of scale efficiency on agricultural NPS pollution, and conducts an empirical test based on China’s statistical data. In the future, there are many directions to be further studied. First, there are many ways to improve scale efficiency. We can move the starting point of research forward and focus on the impact of different scale efficiency improvement methods on agricultural NPS source pollution. Second, explore the conditions under which scale efficiency can inhibit agricultural NPS pollution, and further enrich the toolbox of agricultural NPS pollution treatment. Third, extend the research perspective to farmers at the micro level.

## Figures and Tables

**Table 1 ijerph-18-08798-t001:** Definitions and summary statistics of the variables used in the analysis.

Variable	Definition	Mean	Maximum	Minimum	Standard Deviation
NPS	Levels of agricultural NPS pollution	240.124	859.860	3.221	188.928
Se	Scale efficiency	1.184	7.777	0.107	0.814
Fin	Total agricultural credit scale	3.15 × 10^7^	3.10 × 10^8^	8705	4.90 × 10^7^
Fis	Expenditures of local agriculture, forestry and water affairs	242.756	1310.890	1.250	259.742
W	Disposable income of rural residents	6780.941	30,374.73	1258.000	5048.142
Urb	Ratio of urban population to total population	0.490	0.896	0.220	0.152
Str	Ratio of added value of agricultural industries to the sum of added value of secondary and tertiary industries	0.087	0.283	0.002	0.053

**Table 2 ijerph-18-08798-t002:** Estimation results of panel data model.

Variable	Model		
	Pooled OLS	Fixed Effects	Random Effects
	(1)	(2)	(3)
Constant	197.505 (7.09)	187.382 (10.37) ***	197.505 (7.09) ***
Se	2.901 (1.54)	2.914 (1.61)	2.901 (1.54)
Fin	−5.07 × 10^−7^ (−6.14) ***	−5.26 × 10^−7^ (−6.62) ***	5.07 × 10^−7^ (−6.14) ***
Fis	0.222 (10.95) ***	0.216 (11.01) ***	0.222 (10.95) ***
W	−0.004 (−5.49) ***	−0.004 (−5.58) ***	−0.004 (−5.49) ***
Urb	107.145 (2.96) ***	131.779 (3.73) ***	107.145 (2.96) ***
Str	−258.822 (−4.14) ***	−264.09 (−4.38) ***	−258.82 (−4.14) ***
Adjusted R2	0.467	0.468	0.467
F/Wald	473.2 ***	85.4 ***	473.2 ***
Obs	620	620	620

Note: *** indicates significant levels of significance at 1%. Non-marked is not significant.

**Table 3 ijerph-18-08798-t003:** Estimation results of spatial effect by using SAR model.

Variable	SAR Model					
	(4)	(5)	(6)	(7)	(8)	(9)
Constant	54.087 (1.52)	79.296 (2.25) **	127.289 (3.6) ***	110.488 (3.22) ***	87.663 (2.43) **	124.584 (3.15) ***
Se	3.938 (1.95) *	3.701 (1.87) *	2.823 (1.53)	3.137 (1.77) *	3.118 (1.76) *	3.027 (1.72) *
Fin		2.55 × 10^−7^ (5.8) ***	−4.14 × 10^−7^ (−5.2) ***	−4.8 × 10^−7^ (−6.38) ***	−5.2 × 10^−7^ (−6.8) ***	−4.88 × 10^−7^ (−6.28) ***
Fis			0.154 (9.93) ***	0.23 (12.5) ***	0.228 (12.39) ***	0.216 (11.3) ***
W				−0.005 (−7.03) ***	−0.005 (−7.49) ***	−0.005 (−6.73) ***
Urb					92.262 (2.64) ***	74.265 (2.07) **
Str						−148.335 (−2.33) ***
ρ	0.7152 (17.91) ***	0.585 (11.49) ***	0.336 (5.55) ***	0.468 (7.96) ***	0.4 (6.1) ***	0.337 (4.64) ***
Lgt_theta	−2.998 (−21.88) ***	−2.979 (−21.68) ***	−3.035 (−22.18) ***	−3.055 (−22.34) ***	−3.081 (−22.52) ***	−3.087 (−22.56)
Sigma2	1626.038 (16.97) ***	1573.556 (17.03) ***	1471.582 (17.13) ***	1258.292 (17.09) ***	1247.165 (17.11) ***	1240.055 (17.12) ***
Adjusted R2	0.0084	0.2708	0.3905	0.4264	0.4538	0.4679

Note: *, ** and *** indicate significant levels of significance at 10%, 5% and 1%, respectively. Non-marked is not significant.

**Table 4 ijerph-18-08798-t004:** Estimation results of decomposed spatial effect.

Variable	Direct Effect	Indirect Effect	Total Effect
Se	3.12 (1.71) *	1.613 (1.41)	4.734 (1.66) *
Fin	−4.95 × 10^−7^ (−6.56) ***	−2.56 × 10^−7^ (−2.72) ***	−7.51 × 10^−7^ (−5.23) ***
Fis	0.22 (11.8) ***	0.113 (2.87) ***	0.333 (6.89) ***
W	−0.005 (−7.27) ***	−0.003 (−2.45) **	−0.008 (6.89) ***
Urb	74.807 (2.19) **	37.245 (1.86) *	112.051 (2.21) **
Str	−144.295 (−2.26) **	−70.002 (−2.05) **	−214.297 (−2.33) **

Note: *, ** and *** indicate significant levels of significance at 10%, 5% and 1%, respectively. Non-marked is not significant.

## Data Availability

Data available from the authors upon request.

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
