# Peer review of "Can Increasing Scale Efficiency Curb Agricultural Nonpoint Source Pollution?"

_ijerph, 2021, doi:10.3390/ijerph18168798_

Round 1
Reviewer 1 Report
I suggest that the revision for "Can increasing scale efficiency curb agricultural nonpoint source pollution?" still requires additional improvements in the writing style.
I include a list of suggested format and English grammar changes here. If these authors incorporate some improvements in the revision, then yes, I suggest publishing in Intern. J. Environ. Res. and Public Health.

Author Response
Dear Editor Nikolina and reviewers,
We thank the reviewers for the time and effort that you have put into reviewing the previous version of the manuscript. Their suggestions have enabled us to improve our work. Based on the instructions provided in the review letter, we uploaded the file of the revised manuscript. Accordingly, we have uploaded a copy of the original manuscript with all the changes highlighted by using the track changes mode in MS Word.
Appended to this letter is our point-by-point response to the comments raised by the reviewers.
We would like also to thank you for allowing us to resubmit a revised copy of the manuscript.
Sincerely,
Xiang Gao

This manuscript is a resubmission of an earlier submission. The following is a list of the peer review reports and author responses from that submission.
Round 1
Reviewer 1 Report
The manuscript is on evaluating impacts of scale efficiency on agricultural NPS pollution. It is well written and adequately described. I liked the manuscript in present form so I think it can be accepted in this form.
Strength: well explained rationale, methods, variables and analysis. NPS is evaluated with respect to natural, social and economic variables. Combination of different types of independent variables is the strength in this paper. Also, I liked how the authors have categorized effects into direct and indirect. I feel this paper is a good contribution on presenting a theoretical model depicting impact of scale efficiency on agricultural NPS pollution. Few editorial changes can be
Line 305: give full form of SECH.
Line 383: Table 1, need to write full form of the variables. Figure/Table captions need to be self-explanatory.
Line 481: give full form of SAR in table caption
Author Response
Dear reviewer,
We thank you for the time and effort that you have put into reviewing the previous version of the manuscript. Your suggestions have enabled us to improve our work. Based on the instructions provided in your letter, we uploaded the file of the revised manuscript. Accordingly, we have uploaded a copy of the original manuscript with all the changes highlighted by using the track changes mode in MS Word.
Appended to this letter is our point-by-point response to the comments.
Thank you again.
Sincerely,
Xiang Gao

Reviewer 2 Report
The English and grammatical context of the paper needs to be thoroughly reviewed to enhance the readability of the paper.
Definition of parameters in Eqn 2 should be given (Line 222) even if they are defined elsewhere later in the text.
Author Response

(The authors gave the same response as above.)

Reviewer 3 Report
I suggest in the attached file that this work might be more valuable if published in a journal with an emphasis on agricultural economics and/or policy.

Author Response

(The authors gave the same response as above.)
